# A Novel Time-Sensitive Composite Similarity Model for Multivariate Time-Series Correlation Analysis

**DOI:** 10.3390/e23060731

**Published:** 2021-06-08

**Authors:** Mengxia Liang, Xiaolong Wang, Shaocong Wu

**Affiliations:** 1College of Computer Science and Technology, Harbin Institute of Technology, Harbin 150001, China; liangmengxia@hotmail.com; 2College of Computer Science and Technology, Harbin Institute of Technology, Shenzhen 518055, China; wushaocong2013@gmail.com

**Keywords:** dynamic time warping, time-series segmentation, time-series correlation, temporal features

## Abstract

Finding the correlation between stocks is an effective method for screening and adjusting investment portfolios for investors. One single temporal feature or static nontemporal features are generally used in most studies to measure the similarity between stocks. However, these features are not sufficient to explore phenomena such as price fluctuations similar in shape but unequal in length which may be caused by multiple temporal features. To research stock price volatilities entirely, mining the correlation between stocks should be considered from the point view of multiple features described as time series, including closing price, etc. In this paper, a time-sensitive composite similarity model designed for multivariate time-series correlation analysis based on dynamic time warping is proposed. First, a stock is chosen as the benchmark, and the multivariate time series are segmented by the peaks and troughs time-series segmentation (PTS) algorithm. Second, similar stocks are screened out by similarity. Finally, the rate of rising or falling together between stock pairs is used to verify the proposed model’s effectiveness. Compared with other models, the composite similarity model brings in multiple temporal features and is generalizable for numerical multivariate time series in different fields. The results show that the proposed model is very promising.

## 1. Introduction

With the development of computer technology, artificial intelligence, big data and cloud computing, an increasing number of people are relying on computer algorithms to address problems in all fields. People in different fields attempt to use artificial intelligent technology to make their work simpler, faster and more accurate, especially in finance [1]. Due to the digitization of financial transactions, large amounts of financial data with considerable implicit information are generated and stored. How to use these data to help people invest has become an issue of common concern for people majoring in both computer science and finance. There are quite a few people demanding accurate predictions of financial indicators so that they can adjust their investment portfolio in time to gain more returns or reduce deficits. In fact, stock fluctuation is complicated and difficult to predict accurately, and most of the existing approaches can provide investors with only some efficient advice and cannot always provide returns. However, identifying the correlation between different stocks still has meaning for investors in helping them make investment decisions.

Generally, there are correlations between different stocks that could be reflected in price curves, such as stocks rising or falling together, stocks with one rising and another falling and stocks that are similar in shape but unequal in length fluctuations appearing at different times, which could be abstracted from the situation shown in Figure 1. Identifying the relationship between stocks can provide investors with a very efficient investment reference. For example, if a stock kept going up or had a very positive trend on recent days, its similar stocks may have high probability to go up as well, such as the rising-or-falling-together relationship shown in Figure 1. In contrast, if a stock starts to fall, its similar stocks may also follow suit. This means that if an investment portfolio has some similar declining stocks at the same time, then it may lead to enormous losses. According to the cognition of the similarity of stocks, investors could adjust their investment portfolio in time to obtain more returns or avoid enormous losses.

To explore the relationship between stocks, people begin to focus on the fluctuation of stock prices. Because stock price data are stored as time series, the relationship between different stocks is essentially a composite situation of time-series similarity. Considering that different stocks may have different volatility, as some rise or fall quickly and others slowly, they may take different amounts of time to exhibit similar fluctuations. This leads to the situation in which different stock price sequences are nonaligned on a timeline. Dynamic time warping (DTW) is usually used to compute the distance between similar time series of unequal length. DTW can warp the sequences to align the most similar points and obtain the best matches between points of different sequences; therefore, it is usually used for similar price pattern extraction [2,3,4,5]. In most DTW-based approaches, only one temporal attribute, for example, the price, is usually considered. However, considering the actual situation of the stock market, only taking price sequences into account cannot reflect or represent the entire situation of the stock market, let alone accurately predict trends. Stock features such as turnover rate and earnings ratio, which are sequential, also have their impact and representativeness for a stock. These features could help reflect the global similarity between different stocks. In the same way, one attribute’s similarity could not represent the similarity between stock couples. Therefore, we need a new measurement that considers more temporal features to measure the similarity between different stocks.

In this paper, we propose a DTW-based time-sensitive composite similarity model to estimate the similarity between different stocks to detect the correlation between them and help investors adjust their investment portfolio. The contributions of our research are described as follows.

A time-series segmentation approach that is designed for DTW is proposed. Generally, when using DTW to compute two time-series distances, similar important points such as poles need to be aligned, so we design a segmentation that cuts time series by the number of eligible peaks and troughs and stipulate counting rules to ignore tiny fluctuations.A time-sensitive composite similarity model is proposed that can consider more sequential stock parameters and combine the traditional ‘rise or fall together’ similarity measures to measure the similarity of different stocks.Comparisons of prediction accuracy between traditional similarity measures are devised to validate the effectiveness of the proposed similarity model, and the time sensitivity of the composite model is also embodied in the experimental results.

The remainder of this paper is organized as follows. In Section 2, related work about stock relationships and related applications of DTW are presented. Section 3 describes the time-series segmentation approach and the composite stock similarity model we proposed. Section 4 presents the experimental setup and performance of the approach we proposed, as well as a comparison of the experiments and performances. A discussion about the differences in experimental results is also given in Section 4. Finally, the conclusions and directions for future research are given in Section 5.

## 2. Related Work

### 2.1. Correlation

Research on the correlation [6] between different entities could help improve the efficiency of research targets. For example, we could obtain the composition of distributed resource services by researching the correlation between resource services from different organizations to improve resource utilization [7]. Researchers filtered the feature data of turbine groups at certain distances based on the correlation to optimize the forecasting effect on wind power further by clustering [8]. Majnu et al. show the limitations of two current dynamic correlation estimation approaches and present an alternate approach for dynamic correlation estimation based on a weighted graph [9]. Analyzing the correlation between stock prices and different financial indexes could provide a valuable reference to help investors make long-term investment decisions [10]. Random matrix theory is used to analyze the cross-correlations of price changes of different cryptocurrencies [11]. For more accurate predictions on the stock market, researchers propose researching the correlation between corporations and incorporating the information on the related corporations of a target company [12]. To underscore the potential for using multilayer network tools to study the time-varying correlations of financial assets, the authors of [13] apply recent innovations in network science to analyze how correlations of stock returns evolve over time. A complex network could be used to research stock correlation, so Yan et al. proposed to use part mutual information for developing the stock network [14]. To obtain better portfolio allocation and risk management, researchers began to research the correlation between different stocks [15].

### 2.2. Stock Time-Series Correlation

Today, more researchers and investors have begun to focus on the technical analysis of the stock market [16,17]. When talking about the correlation between stocks, some research changes stock similarity to the graphic similarity of patterns [18], calculating the distance of the price vector and classifying patterns to identify predictive stock patterns. Wang [19] constructed a Pearson-correlation-based network and a partial-correlation-based network to analyze the correlation structure and evolution of world stock markets. Guan [20] proposed a forecasting model based on neutrosophic logical relationships and employed a Jaccard similarity measure to find the most proper logical rule for forecasting. Xi [21] created a stock-associated network model based on financial indicators and explored the structural similarity of financial indicators of stocks. Zhang [22] defined an intracoupled attribute value similarity and an intercoupled attribute value similarity to construct a stock correlation matrix to assist in tensor decomposition. Most of these stock similarity studies are based on the static features of stocks, but many stocks’ features are temporal and dynamic, so we decided to define a dynamic similarity of stocks using dynamic features. Then, our research has turned to DTW, which is good at calculating the distance between time series of different lengths.

### 2.3. Dynamic Time Warping

DTW was first proposed and applied to spoken word recognition in 1978 [2] and has been used in pattern recognition [23], time-series data processing [24,25,26,27], signature verification [28,29], speech segment clustering [30], exceptional motion capture [31], etc. It was first used to obtain the optimal alignment between points in both template sequences and test sequences, calculating the distance to obtain two sequences aligned and judge whether two sequences are similar. Currently, DTW is widely used and modified as a similarity calculation method [32,33]. Because it is good at aligning the most similar points and obtaining the distance between two similar time series, many researchers have applied it to the recognition of similar stock patterns [3,4]. Tsinaslanidis [5] proposed an algorithmic approach using mainly the DTW algorithm and two of its modifications, subsequence DTW and derivative DTW, to capture common characteristics for helping stocks’ bullish and bearish class predictions.

The process of the DTW algorithm is shown below.

**Definition** **1.** **Dynamic time warping**. *Given two sequences, X = (x_1_, x_2_, … x_m_) and Y = (y_1_, y_2_..., y_n_), the distance function of any point-to-point in two sequences is d(i, j) = f(x_i_, y_j_) ≥ 0. Due to m ≠ n, an m × n matrix is constructed to obtain two aligned sequences. To obtain the aligned matrix, a sequence distance matrix D is obtained, whose rows correspond to sequence X; columns correspond to sequence Y, and the element of matrix D(i, j) represents the distance from x_i_ to y_j_, which is d(x_i_, y_j_). Generally, Euclidean distance is used as the distance function. Then, the loss matrix Dc is obtained via the following steps:**Step 1*: *Set**D_c_*(1, 1) = *D*(1, 1);*Step 2*: *D_c_*(*i*, *j*) = *MIN*(*D_c_*(*I* − 1, *j* − 1), *D_c_*(*I* − 1, *j*), *D_c_*(*i*, *j* − 1)) + *D*(*i*, *j*).

Follow the two steps, and repeat Step 2 until the element in the last row and last column is obtained, which is also the DTW distance between two sequences.

To illustrate the process of obtaining the DTW distance better, we use an easy example to show how DTW works. There is an original time series *S_0_* = {3,6,8,5,7,2}, another time series *S_1_* = {2,6,7,5,6,7,2,1}; the curves of the two time series are shown in Figure 2. To obtain the DTW distance between two time series, we choose distance function *d*(*i, j*) = *f*(*x_i_*, *y_j_*) = |*x_i_* − *y_j_*|. Then, the distance matrix is obtained which is shown in (a) of Figure 3. Follow the steps obtaining the loss matrix described in Definition 1, we obtain the loss matrix which is shown in (b) of Figure 3.

We could obtain a warping path through the loss matrix if needed. The least-cost path from the first element of matrix, which is the first row and first column located, to the last element, which is also the last row and last column located, is the warping path between two time series. The warping path obtained from loss matrix is shown in (a) of Figure 4. The value of the last element is also the DTW distance which we use in this paper. According to the warping path, we obtained pairs of points which could be aligned to each other in (b) of Figure 4, and two points connected by the red dotted line could be aligned.

When using DTW to compute the distance between two time series, there are some constraints here:Monotonicity. All the points in the time series should be aligned by the time order. For example, in (a) of Figure 5, all the black dotted lines connect all the pairs of points aligned with each other, but the pair of points connected by the red dotted line is not allowed to be aligned.Continuity. To ensure that all the points in the two sequences are matched in the calculation process, the calculation of the two points’ distances cannot be skipped, and it should be continuously calculated. It is easy to find that *D_c_*(*i*, *j*) is dependent on *D_c_*(*i* − 1, *j* − 1), *D_c_*(*i* − 1, *j*) and *D_c_*(*i*, *j* − 1) in step 2 of the loss-matrix-obtaining process. For example, if we skip the calculation of *D_c_*(*i* − 1, *j*), that means we could not obtain the value of *D_c_*(*i* − 1, *j*); then, the value of *MIN*(*D_c_*(*i* − 1, *j* − 1), *D_c_*(*i* − 1, *j*), *D_c_*(*i*, *j* − 1)) could not be obtained, and that means that the value of *D_c_*(*i*, *j*) could not be obtained.Boundary conditions. The start point and end point of one time series should be aligned with the start point and end point of another time series. When matching one time series to another, the matching direction should be consistent, both from the start point to the end point. For example, in (b) of the Figure 5, if we want to obtain the DTW distance between two time series, the pairs of points connected by the red dotted lines must be aligned with each other.

In addition, there are some constraints that could also be added in practical application:Slope constraints. To avoid the same points in one time series being aligned too many times in another time series, just as in (c) in Figure 5, the slope could be constrained.Warping windows. Generally, the best-matching paths tend to be near the diagonal, just as in the condition in Figure 6, so sometimes only a suitable path in a window near the diagonal needs to be considered.

## 3. DTW-Based Temporal Composite Similarity Model

To find the correlation between entities of financial time series, a time-sensitive composite similarity model designed for multivariate time-series correlation analysis based on dynamic time warping is proposed. Related definitions and algorithms are described in this section.

### 3.1. Peaks and Troughs Time-Series Segmentation (PTS)

DTW was originally developed for similar but unequal length speech recognition. Similar but unequal length time series may be the same word’s speech. Therefore, DTW is good at recognizing the similarity between time series that are similar but unaligned in the timeline. However, DTW will cause alignment mistakes due to local noise in the time series. To overcome the impact of local noise on DTW applications while following strict boundary conditions, we propose the PTS approach to cut the time series of different stocks’ temporal features to ensure that all the time-series samples will have the same number of fluctuations.

**Definition** **2.** **Peak and trough**. *A time series instance T = [v_1_, v_2_, …, v_k_], v_k_* ∈ ℝ *. Set a random point in T, denoted by T[x] = v_x_, x* ∈ *[1,k]. If any other point T[y] = v_y_, y ≠ x is adjacent to T[x], and T[y] ≤ T[x] exists for all T[y], then T[x] is a peak point P_p_ (red points in Figure 7). In contrast, if T[y] > T[x] exists for all T[y], then T[x] is a trough point P_t_ (green points in Figure 7).*

The peak and trough points were extracted to divide the fluctuation of the time series. A definition of an eligible fluctuation is as follows:

**Definition** **3.** **Fluctuation**. *Given a time series instance T = [v_1_, v_2_, …, v_k_], v_k_* ∈ ℝ*, its turning point collection*
P^={Pp,Pt}
*is a set of all peak points and trough points. The difference between a peak point P_a_ and a trough point P_b_ in P^ is D_a,b_, Da,b=|va−vb|(Da,b∈ℝ). If D_a,b_ is greater than or equal to the given constant δ, the subsequence between P_a_ and P_b_ is considered an eligible fluctuation F_a,b_.*

Different constants δ will lead to completely different divisions of fluctuations in the same time series. As shown in Figure 8, the instance is divided into two eligible fluctuations (*F*_1,5_ and *F*_5,30_) when δ = 0.5, but if δ = 1.0, the same instance will be divided into two eligible fluctuations (*F*_1,20_ and *F*_20,30_).

**Definition** **4.** **PTS (Peaks and Troughs Time-series Segmentation)**. *Given the input sequence X = [x_1_ … x_m_] with length m, for convenience, X [i] is used to represent the i^th^ element in sequence X, X [i] = x_i_, 0 ≤ i ≤ m. A peak and trough deviation threshold value δ is used to judge whether the subsequence is an eligible fluctuation, and the number of eligible fluctuations n is used to find the split point and controls the length of segmentations. We can split the sequence X as follows:**Step 1: Backtracking from the last point (X [m]) of X and finding the first peak as the beginning point of it, call this point X [start]*.*Step 2: Backtracking X from the point X [start], we found in the last step, X [i]. If the last step is step 1, X [i] may be X [start]. If the last step is step 2, X [i] may be any point before X [start]. Find a trough next to X [i] and set it as X [j]. If the deviation of the value of X [i] and the value of X [j] is larger than the threshold value δ, similar to the peak and trough deviation marked by the red line in Figure 9, then we take this peak and trough as eligible fluctuations. If the deviation of the value of X [i] and the value of X [j] is not larger than the threshold value δ, similar to the peak and trough deviation marked by the blue line in Figure 9, then we go on to backtrack and find the next trough that could meet the condition.**Step 3: Go on the backtrack sequence to find a new peak next to the trough, which is obtained in step 2, and repeat step 2 until the number of eligible fluctuations reaches n_ef_ (the eligible fluctuation number set according to segmenting demand). We suppose the last peak we find is X [end], where X [end…start] is the sequence used as input to the DTW approach.*

The entire PTS process can also be described by Algorithm 1. PTS could ignore the tiny fluctuation when cutting the sequences by tuning the threshold value δ so that only obvious fluctuations could be the basis of the cutting approach, which could obtain more accurate similar sequences for DTW. Figure 10 shows the comparison between the original time series and the target time series cut by peak and trough segmentation with different δ and *n_ef_*.

The PTS has two parameters; δ could be decided by the user’s psychological anticipation of minor fluctuations that the user wants to ignore. For example, in the financial market, it could be the psychological endurance range. At the experimental level, δ also controls the granularity and avoids time series with a large granularity gap matching with each other. The parameter *n_ef_* is used to control the number of the eligible fluctuations. Both parameters control the length of history data which is used to analyze the correlation. Because the correlation is changeable in different periods, the length of history data should be in the proper range. Generally, the values of two parameters are adjusted through the experiment results; the process of adjusting parameters is illustrated in the Section 4.2.
**Algorithm 1** Algorithm for PTS**Input:***Sequence*,    ▷ The original time series      δ,   ▷ Peak and trough deviation threshold value      *n_ef_* ▷ The number of eligible fluctuations**Output:**
*newSequence**fluctuation* = 0*i* = len(*Sequence*) – 1*j* = 0**while** *i* >= 2 **do***i* = *i* – 1**if** *Sequence*[*i* − 1] <= *Sequence*[*i*] and *Sequence*[*i*] >= *Sequence*[*i* + 1] **do**  ▷find a peak  *j* = *i*  **while** *j* >= 2 **do**  *j* = *j* – 1 **if** *Sequence*[*j* − 1] >= *Sequence*[*j*] and *Sequence*[*j*] <= *Sequence*[*j* + 1] and abs(*Sequence*[*i*] − *Sequence*[*j*]) >= δ **do** ▷ find a trough which could construct an eligible fluctuation with the peak we found before  *fluctuation* = *fluctuation* + 1  *i* = *j*  **break**  **end if**  **end while****end if****if**
*fluctuation* == *n_ef_* **do**  **break****end if****end while***newSequence* ← *Sequence*[*j:*]

### 3.2. Time-Sensitive Composite Similarity Model

When we refer to the similarity of two stocks, the most intuitive expression is that if they ‘rise or fall together (*roft*)’ frequently, then they are more likely to be similar. Therefore, we take the number of days rising or falling together in the same period of time as one attribute of similarity, which is one of the traditional measures of stock correlation. Obviously, the similarity of two stocks and the number of days rising or falling together are proportional. The number of days of two stocks rising or falling together can be calculated by the sequential data of stock change, and the process is described by Algorithm 2.

However, the number of days rising or falling together is not enough to represent all similar situations. For example, if one stock had risen 3 days and then fell, but another similar stock began to rise 2 days later than the first one and rose 3 or more days and then fell, then the number of days rising or falling together may be only 1, but the whole trend curve is not only similar at one day, similar to the situation shown on the left side of Figure 11. The two lines have very similar trend curves, but they are not aligned on the timeline. We need to use DTW to align the most similar point and compute the distance between the two similar lines, which is shown on the right side of Figure 11.
**Algorithm 2** Calculation of number of days rising or falling together**Input:** *changesequence1*, *changesequence2* ▷change sequences of stock1 and stock2**Output:***roft*            ▷number of days rising or falling together*roft* = 0**if** len(*changesequence1*) > len(*changesequence2*) **do** *days* = len(*changesequence2*)**end if****else do** *days* = len(*changesequence1*)**end else****for** *i* = 0 to *days* − 1 **do****if** *sequence1*[*i*] > 0 **do**   **if** *sequence2*[*i*] > 0 **do**   *roft* = *roft* + 1   **end if****end if****if** *sequence1*[*i*] < 0 **do**   **if**
*sequence2*[*i*] < 0 **do**   *roft* = *roft* + 1   ****end if******end if****if**
*sequence1*[*i*] == 0 **do**   **if**
*sequence2*[*i*] == 0 **do**   *roft* = *roft* + 1   **end if****end if****end for**

Generally, we use the closing price to analyze the price trend or predict the future stock price, but the stock market is very complex, and considering only the closing price cannot reflect the entire situation of a stock. It is very difficult to find the real relationship using only one attribute; there are also many other sequential features that influence stock price trends, and we can see that curves of different features of the same stock can be very different, which is shown in Figure 12. Examples of daily raw stock data are shown in Table 1.

Only one feature’s DTW distance could not represent the similarity of two stocks, so we decided to combine more features’ DTW distances and their rise-or-fall-together times to obtain a composite similarity to compute the similarity between two stocks. Because the similarity is proportional to the number of rises or falls together and inversely proportional to the DTW distance, we define the similarity as follows (1):(1)$$Similarity=roft\overset{n}{\underset{i=1}{\sum }}\frac{\lambda_{i}}{DTW\left(feature_{i}\right)+1}$$
In Formula (1), *roft* is the number of days rising or falling together in the same period of time; $\lambda_{1}$…$\lambda_{n}$ are the weights of different sequential features in the similarity; *λ*_1_ + …+*λ_n_* = 1, DTW(*feature_1_*) … DTW(*feature_n_*) are the DTW distances between the target stock and benchmark stock with the sequences of different temporal features (*feature_1_* … *feature_n_*). DTW distances in the composite similarity model are only used to describe the degree of similarity of different temporal features; the matching path between sequences is not considered in this model.

In this similarity formula, we could choose different sequential features of stocks to combine and use λ to tune the weight of each feature in the composite similarity to make the similarity closer to reality.

The whole process of obtaining the composite similarity is shown in Figure 13. The similarity obtained from the composite similarity model is a relative value that is used to compare the similarity between stocks similar to the benchmark stock. Only one single stock’s similarity value is meaningless, and it is meaningful when compared with other similar stocks’ similarity values. If one stock’s similarity is larger than that of another stock, then this stock is more similar to the benchmark stock than to another stock. We will evaluate the similarity model and compare it with other similarity measures in the next section.

## 4. Performance Evaluation

In this section, the DTW-based composite similarity model proposed in the previous section is applied to a stock database containing the basic daily information of 300 CSI stocks collected from Tushare Pro (https://tushare.pro/, accessed on 1 February 2021). We first introduce our dataset and experimental settings. Then, we analyze the outputs of similarity computing. Finally, we compare the result of this model with the similarity calculated only by DTW of the closing price and similarity calculated by the rising-or-falling-together number.

### 4.1. Experimental Setup

For evaluation, the stock data we use are the stocks in the CSI 300 Index, whose samples are selected from the Shanghai and Shenzhen stock markets, cover most of the market capitalization and can reflect the income of mainstream investment in the market. We use all stocks in CSI 300 and collect their basic daily information, including the closing price, turnover rate, volume ratio, price-to-earnings (PE) ratio, price-to-earnings trailing twelve months (PETTM) ratio, price-to-book (PB) ratio, price-to-sales (PS) ratio and price-to-sales trailing twelve months (PSTTM) ratio, as the features to compute the composite similarity. Part of the stock list is shown in Table 2. As there are too many columns in a grid of stock quotation data, some of them are shown in Table 3. Stock data from the date 1 January 2018 to the date 31 December 2018 set as Group 1 and data from the date 1 January 2019 to the date 31 December 2019 set as Group 2 are used to compute the similarity. The data from 2 January 2019 and 2 January 2020 are used to verify whether the similar stocks obtained from the composite similarity model rise or fall together with the stock we choose as the benchmark.

The stock of Sinopec, whose stock code is ‘600028. SH’, is chosen as the benchmark; it is just a case to show how the model works, and certainly any other stock can be chosen as the benchmark according to investment preference. The composite similarities of the other 299 stocks in CSI 300 are computed by the composite similarity model we used.

For comparison, we also use the number of days rising or falling together in 2019 and the DTW of the stock closing price as the similarity of stocks. The similarities of stocks are obtained from the composite similarity model and these two methods, and we compare the rise-or-fall-together rate after computing the similarity to see whether the model is efficient.

The rise-or-fall-together rate is obtained from Formula (2):(2)$$roftrate=\frac{num\left(roft_{t1}\cap^{\;}roft_{t2}\cap^{\;}\ldots \ldots \cap^{\;}roft_{tn}\right)}{n_{s}}$$
In Formula (2), *roftrate* is the rise-or-fall-together rate; *roft_t1_* … *roft_tn_* is the set of stocks whose prices that rise or fall together with the benchmark stock on day *t_1_* … *t_n_*, *num*(), which is the method that obtains the number of stocks in a stock set; *n_s_* is the number of samples we choose to compute the rise-or-fall-together rate. When *n_s_* is 299, we obtain the average rise-or-fall-together rate of the whole sample.

### 4.2. Results and Discussion

In Group 1, we set δ = 0.3 in the PTS to ignore minor fluctuations and *n_ef_* = 10 to ensure that all the sequences have a similar number of eligible peaks and troughs and that the lengths of sequences are the right size. We set the weight of eight features, closing price, turnover rate, volume ratio, PE ratio, PETTM, PB, PS and PSTTM, as (0.1,0.1,0.3,0.1,0.1,0.1,0.1,0.1), with which we experimented many times to obtain preferable results. After cutting the sequences and computing the composite similarity, 299 stocks’ similar degrees with the stock ‘600028.SH’ are obtained and sorted in descending order. The top ten similarity results are shown in Table 4, and the bottom ten similarity results are shown in Table 5.

Similarity calculated only by the DTW distance of the closing price and similarity calculated by the number of days rising or falling together are used to compare the experimental results. On the transverse side, we compare the rising-or-falling-together rate in the top 100 and top 150 similar stocks. The results are shown in Table 6 and Figure 14. Longitudinally, we compare the top 100 rising or falling together rates at 1 day, 2 days, and 3 days after computing the similarity, which is shown in Table 7 and Figure 15.

The stocks are sorted by similarity in descending order, and stocks on the top are the most similar stocks to the benchmark stock. If the benchmark stock’s trend is used to predict the trend of similar stocks, then the prediction accuracy, which is also the rise-or-fall-together rate, could reach 83% in the top 100 similar stocks and 75.3% in the top 150 similar stocks, whose similarity is obtained by the composite similarity model. This result is better than 76% in the top 100 and 74.6% in the top 150, whose similarity is obtained by the number of days rising or falling together. Both of these results are higher than the average rise-or-fall-together rate of the whole sample. The composite similarity was 7.1% better than the average rate of the whole sample. However, if only the DTW distance of the closing price is chosen as the similarity measure, then the rise-or-fall-together rate is lower than the average rate of the whole sample.

In the financial market, there is a time difference between ‘buy’ and ‘sell’ investment behavior, for example, the ‘T+1’ trading rule in the Chinese stock market, which means that the stock you buy on day T could only be sold out on day T+1. This reality makes a situation in which the prediction for a continuous period of time is also worthwhile. Therefore, whether the similarity obtained by different models will last for a few days is also taken into account. Table 7 shows that if the stocks are sorted by the similarity calculated by the composite model in descending order, then the rate of the same trend could reach 83% on the first day in the top 100 stocks, 50% on the second day and 49% on the third day after calculating the similarity. If the number of days rising or falling together is taken as the similarity, then the rate of the same trend in the top 100 similar stocks could reach only 76% on the first day, and all the rates are lower than those of the composite model and higher than the average rate of the whole sample. As days passed by, all three measures’ rates of the same trend decreased but remained above the average rate. Only DTW distances of close price time series are lower than the average rate, which may be because the fluctuations of stocks in this time period are complex and affected by multiple features. Only one feature’s DTW distance could not cluster similar stocks well, so the top 100 and top 150 similar stocks could not gain a better rise-or-fall-together rate than the average rise-or-fall-together rate. However, the composite similarity measure is always above the other two measures and the average rate of the whole sample over time.

The longitudinal comparison results show that on the first predicting day, both the composite similarity and the similarity measure of the number of days rising or falling together could obtain higher accuracy than the overall average same trend rate, and the composite similarity could obtain more accurate predictions than the other two traditional similarity measures.

The horizontal comparison results show that the composite similarity could obtain a more accurate prediction than the other similarity measures not only on the first prediction day but also on the second and third prediction days. This means that the composite similarity is more time sensitive. In fact, the relationship between different stocks is continuous, which supports researchers using the stock correlation to predict stock trends.

In Group 1, although the values of *δ* and *n_ef_* in the PTS are set as the same for all the features roughly, we still take a few experiments to obtain the appropriate values which could obtain good experiment results. We found that if *n_ef_* is too large, PTS would lose effect, and the time series would not be segmented. We tried to retain as many eligible fluctuations as possible in the time period. After observing the range of data, we finally took *n_ef_* = 10. The value of *δ* is adjusted through the experiment results; that is, the top 100 rising-or-falling-together rates at 1 day, 2 days and 3 days. The experiment results vary with the value of *δ* as shown in Table 8, and the weights of eight features are (0.1,0.1,0.3,0.1,0.1,0.1,0.1,0.1).

The variation in experimental results with the value of *δ* when *n_ef_* = 10 is shown in the Figure 16. It is easy to see that as the value of *δ* goes up, the experimental results of 1 day goes up with mild concussions, peaks at *δ =* 0.30 approximately, then falls. So, we set *n_ef_* = 10 and *δ =* 0.30 in the experiment.

The weights of eight features in Group 1 were not chosen only based on subjective considerations; they were adjusted by the experimental results. Firstly, we took each of the features’ weight as 1/8, and the results were not satisfactory. Then, we thought that maybe different features master the time-series fluctuation in different periods, and we also wanted to find the most effective feature in a period, so we tried up-weighting one feature and checked if it would improve accuracy. Different weights of features and the experimental results of the top 100 rising-or-falling-together rates at 1 day, 2 days and 3 days are shown in Table 9, and in these experiments, *n_ef_* = 10, and *δ =* 0.30.

The weights of eight features which could obtain the best accuracy were chosen. The reason why we choose eight features in this composite model is that the fluctuations of multivariate time series and the correlation between multivariate time series are affected by multiple features; only one feature’s similarity could not reflect the similarity between the entities of multivariate time series. The number of features is not fixed, but we decided upon the number of temporal features that may be related to the target relationship that we want to analyze. Eight is not a threshold value; theoretically, any multivariate time series which have more than one temporal feature could use our composite model to find similar entities. To verify the generality of our model, experiments will be carried out on the other dataset in Section 4.3.

Because different features are in different ranges, we decided to segment different features separately. In Group 2, we set *δ* and *n_ef_* separately for different features to reach a more accurate rise-or-fall-together rate, which is shown in Table 10. According to these variables’ values, the situation of the benchmark stock’s time series before and after cutting is shown in Figure 17. The weight of eight features, closing price, turnover rate, volume ratio, PE ratio, PETTM, PB, PS and PSTTM, are set as (0.1,0.1,0.1,0.1,0.3,0.1,0.1,0.1). This time, we obtained better experimental results on 2020.01.02, which are shown in Table 11.

In Group 2, the composite similarity still achieved the best rise-or-fall-together rate and achieved 9.4% better than the average rate of the whole sample on the rise-or-fall-together rate; it is also better than 7.1% in Group 1 which means that through tuning variables’ values in PTS proposed in this paper could help to obtain a more accurate rise-or-fall-together rate.

All experiments in two groups show that the composite similarity model can effectively cluster similar stocks together through a time-series correlation analysis to help investors adjust their portfolios.

### 4.3. Verification of Generality

To verify the generality, we tested our model on real weather data of 168 Chinese cities collected by AkShare (https://akshare.xyz/, accessed on 24 April 2021). The structure of daily weather data is shown in Table 12. The temporal features chosen for the experiment and their meanings are shown in Table 13.

In the financial market, we care about the price of stocks, so we use the closing price to compute the *roft* in Formula (1) and *roftrate* in Formula (2); in these weather data, we chose the feature temp. The city named Beijing was chosen as the benchmark. Daily weather data from 2020.01.01 to 2020.12.31 were used to compute the composite similarity; the rising-or-falling-together rate of temperatures in the top 50 similar cities on 2021.01.01 was used to verify whether similar cities’ temperature will change consistently with the benchmark city of Beijing. In this experiment, we set *n_ef_* = 30 and *δ =* 1. The weights of features, PM 2.5, PM 10, No_2_, CO, O_3_, SO_2_ temperature, humidity, were set as (0.1,0.1,0.1,0.3,0.1,0.1,0.1,0.1). The traditional similarity measures, the pure DTW distance of temperatures and the number of temperatures rising or falling together were also chosen as the comparison methods.

The top ten cities similar with Beijing are shown in Table 14. The rising-or-falling-together rate on 2021.01.01 in the top 50 similar cities is shown in Table 15.

It is obvious that the composite similarity could achieve better results than other similar methods in weather data. The experiment on weather data verified that the proposed composite similarity model is efficient not only with financial multivariate time-series data but also with other multivariate time-series data.

## 5. Conclusions

In this paper, we studied the correlation between stocks to provide helpful references for investors adjusting investment portfolios and proposed a composite similarity model that composited many different sequential features of stocks. Then, the composite model was compared with other similarity-computing methods to verify its effectiveness and practicability. The results show that the composite model could obtain more accurate clusters than many traditional similarity measures. When adjusting investment portfolios, investors could take an uptrending stock as a benchmark to buy in similar stocks, and when a stock’s price is going down, investors could sell similar stocks in their portfolios. The composite similarity model could help investors find similar stocks according to historical data and adjust portfolios quickly. The model could also be used to find the most effective feature which masters the fluctuation in a period. Experiments on other datasets also proved that the composite similarity model could be used to research multivariate time series in different fields. However, only eight usual temporal features were used in the composite model. Finding more useful stock features, tuning the weights of different features to reach more accurate results and finding other functional forms to describe the relationship between time series’ similarity and temporal features could be directions for future research.

## Figures and Tables

**Figure 1 entropy-23-00731-f001:**
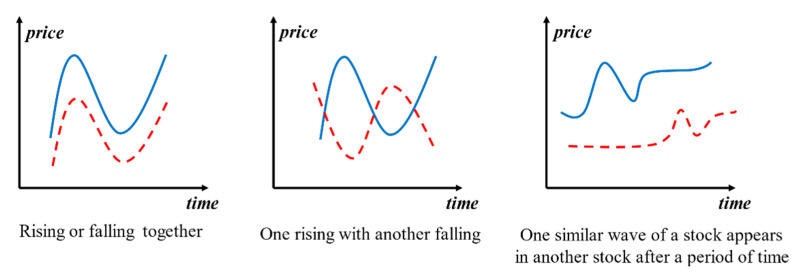
Relationships between two stock price series.

**Figure 2 entropy-23-00731-f002:**
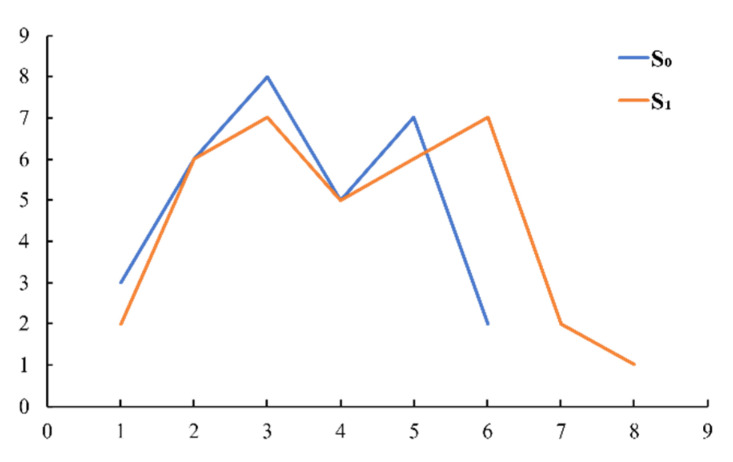
Curves of two time series.

**Figure 3 entropy-23-00731-f003:**
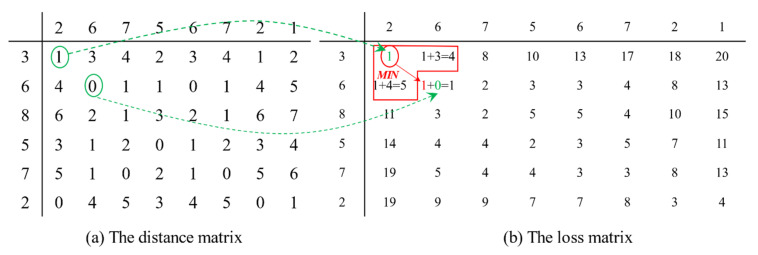
The distance matrix and the loss matrix of two time series.

**Figure 4 entropy-23-00731-f004:**
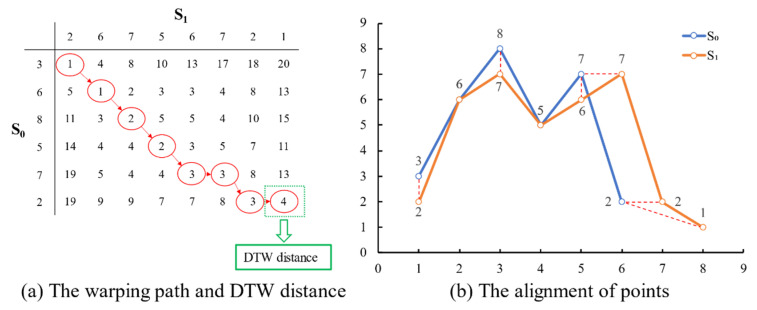
The warping path and the alignment between two time series.

**Figure 5 entropy-23-00731-f005:**
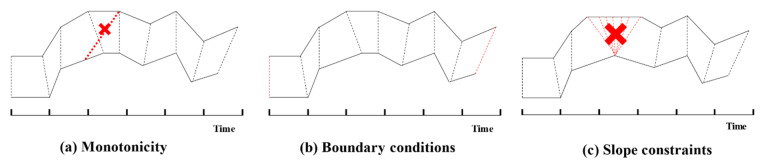
Example of monotonicity, boundary conditions and slope constraints.

**Figure 6 entropy-23-00731-f006:**
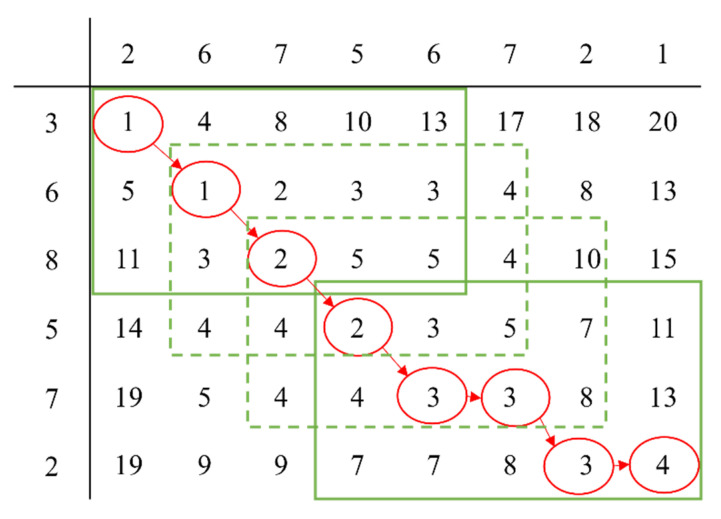
Example of warping windows.

**Figure 7 entropy-23-00731-f007:**
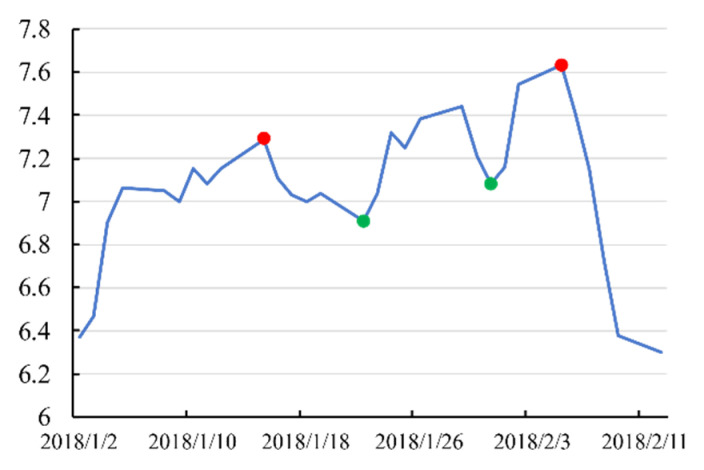
An example of peak and trough points in stock time series. This time-series instance is the close price sequence of stock (600028. SH) in the Chinese market from 2018-01-02 to 2018-02-12.

**Figure 8 entropy-23-00731-f008:**
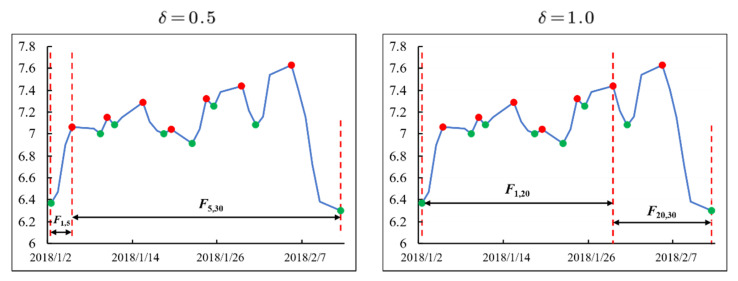
Fluctuation divided by different constants δ.

**Figure 9 entropy-23-00731-f009:**
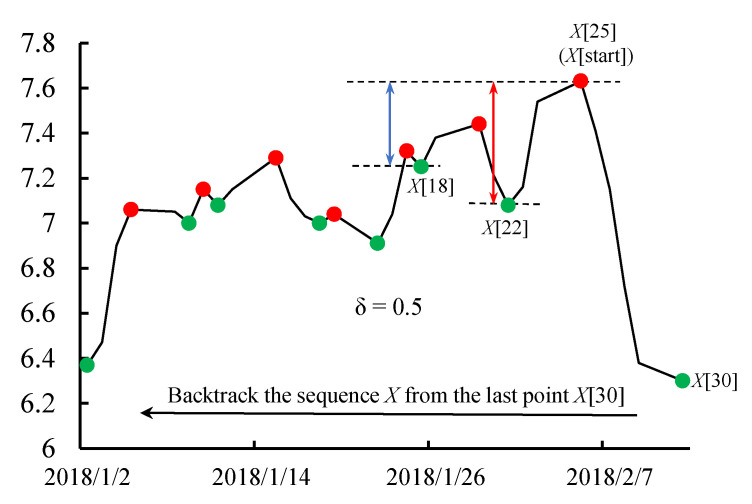
Schematic diagram of PTS when δ = 0.5.

**Figure 10 entropy-23-00731-f010:**
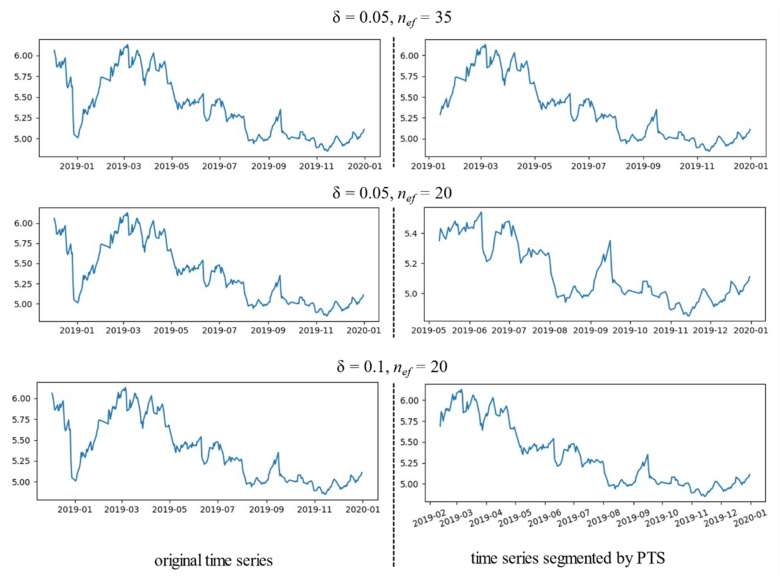
Example of time series segmented by PTS.

**Figure 11 entropy-23-00731-f011:**
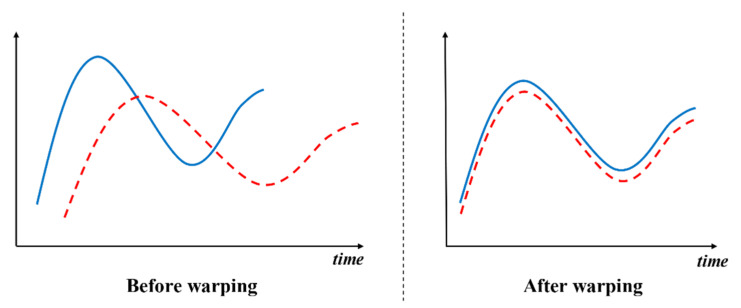
Example of warping different curves to align them.

**Figure 12 entropy-23-00731-f012:**
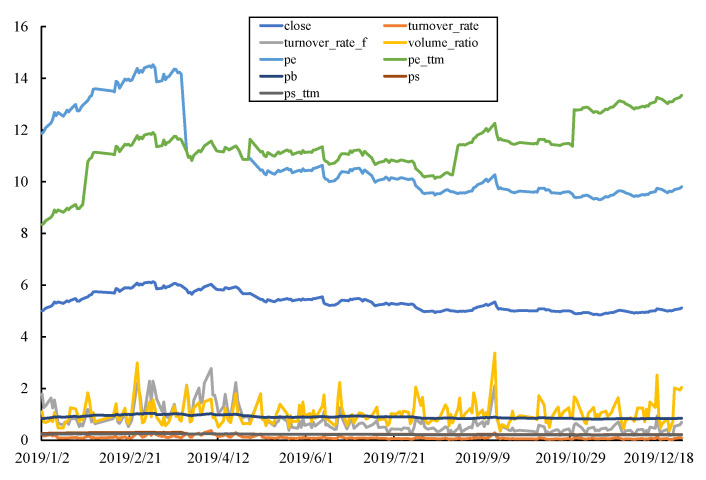
Curves of different features of the same stock.

**Figure 13 entropy-23-00731-f013:**
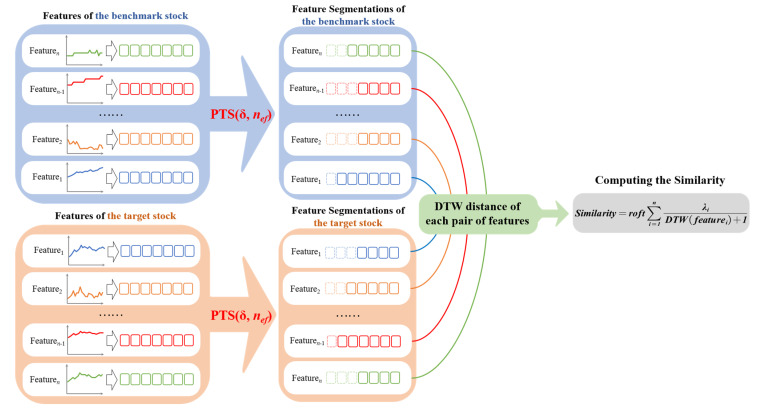
Process of obtaining the composite similarity.

**Figure 14 entropy-23-00731-f014:**
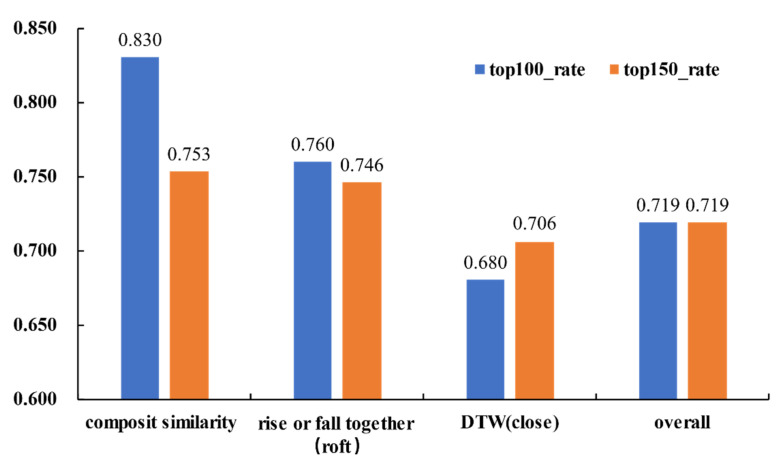
Longitudinal comparison results of Group 1.

**Figure 15 entropy-23-00731-f015:**
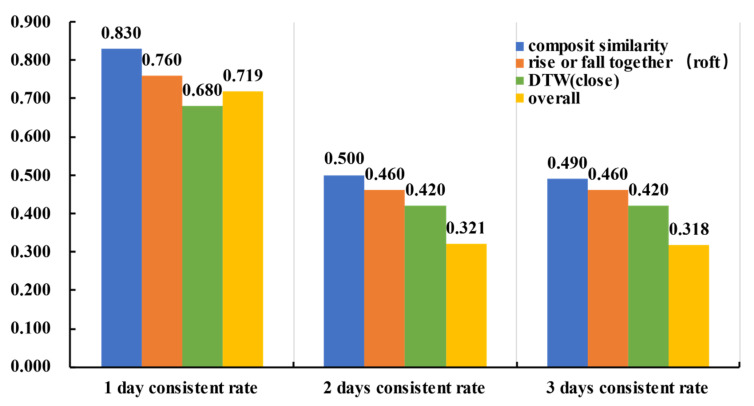
Horizontal comparison results of Group 1.

**Figure 16 entropy-23-00731-f016:**
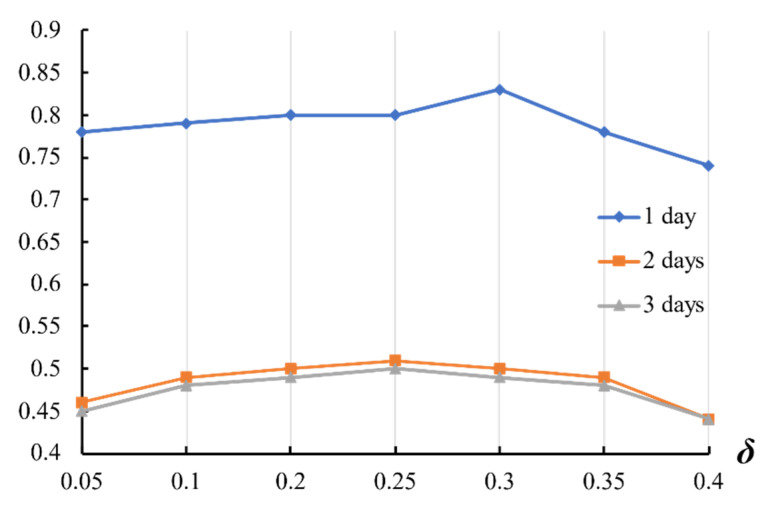
Correlation between experimental results and the value of *δ*.

**Figure 17 entropy-23-00731-f017:**
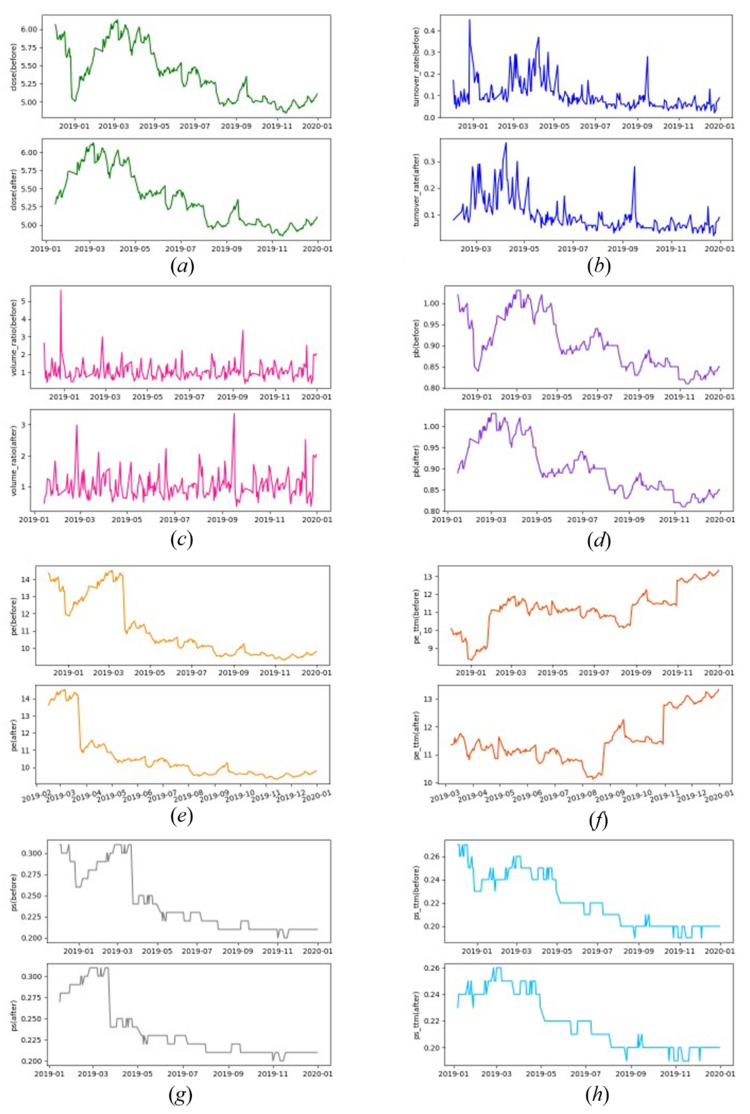
Situation of time series cut by PTS using variable values in Table 8: (**a**) closing price, (**b**) turnover rate, (**c**) volume ratio, (**d**) PB, (**e**) PE, (**f**) PETTM, (**g**) PS, (**h**) PSTTM.

**Table 1 entropy-23-00731-t001:** Example of daily raw stock data.

Feature Name	Description	Value
stockcode	code of a stock	600028.SH
tradedate	trading date	2020-01-02
close	closing price of a day	5.170
turnover_rate	turnover rate	0.130
volume_ratio	volume ratio	2.200
pe	price-to-earnings ratio	9.922
pe_ttm	price-to-earnings trailing twelve months ratio	13.49
pb	price-to-book ratio (total market value/net assets)	0.860
ps	price-to-sales ratio	0.220
ps_ttm	price-to-sales trailing twelve months ratio	0.210

**Table 2 entropy-23-00731-t002:** Some constituent stocks of CSI 300.

No.	Stock Code	Stock Name
1	600000.SH	Shanghai Pudong Development Bank
2	600004.SH	Baiyun Airport
3	600009.SH	Shanghai Airport
4	600010.SH	Inner Mongolia Baotou Steel Union
5	600011.SH	Huaneng Power International
6	600015.SH	Hua Xia Bank
7	600016.SH	China Minsheng Banking Corp., Ltd.
8	600018.SH	Shanghai International Port (Group) Co., Ltd.
9	600019.SH	Baoshan Iron & Steel
10	600023.SH	Zheneng Electric Power Co., Ltd.
11	600025.SH	Huaneng Hydropower
12	600027.SH	Huadian Power International
13	600028.SH	Sinopec
14	600029.SH	China Southern Airlines
15	600030.SH	Citic Securities

**Table 3 entropy-23-00731-t003:** Raw day-by-day stock data. (from 6 June 2019 to 31 December 2019).

Statedate	Stockcode	Close	Turnover_Rate	Volume_Ratio	PE	(PETTM, PB, PS)	PSTTM
6 June 2019	600028	5.480	0.070	0.850	10.516	…	0.22
10 June 2019	600028	5.540	0.100	1.310	10.632	…	0.22
11 June 2019	600028	5.300	0.150	1.750	10.171	…	0.21
12 June 2019	600028	5.260	0.080	0.810	10.094	…	0.21
13 June 2019	600028	5.250	0.080	0.820	10.075	…	0.21
…	…	…	…	…	…	…	…
31 December 2019	600028	5.110	0.090	2.040	9.806	…	0.20

**Table 4 entropy-23-00731-t004:** Top 10 of composite similarity results.

No.	Stockcode	Composite Similarity
1	601618	12.1626
2	600606	11.4819
3	600068	11.1654
4	601669	11.0645
5	000630	11.0280
6	600362	8.6147
7	000709	8.0107
8	600297	7.9419
9	000100	7.0628
10	600104	6.4541

**Table 5 entropy-23-00731-t005:** Bottom 10 of composite similarity results.

No.	Stockcode	Composite Similarity
1	603259	0.9996
2	601066	0.9260
3	600518	0.9106
4	600196	0.8926
5	603986	0.7938
6	002602	0.7108
7	601828	0.6393
8	002450	0.4152
9	002411	0.3485
10	002252	0.1839

**Table 6 entropy-23-00731-t006:** Longitudinal comparison results of Group 1.

Type	Top 100	Top 150
composite similarity	0.830	0.753
number of days rising or falling together	0.760	0.746
DTW (close)	0.680	0.706
average rate of whole sample	0.719	

**Table 7 entropy-23-00731-t007:** Horizontal comparison results of Group 1.

Type	1 Day	2 Days	3 Days
composite similarity	0.830	0.500	0.490
number of days rising or falling together	0.760	0.460	0.460
DTW (close)	0.680	0.420	0.420
average rate of whole sample	0.719	0.321	0.318

**Table 8 entropy-23-00731-t008:** Different variables’ values and related experimental results.

Value of *n_ef_*	Value of *δ*	1 Day	2 Days	3 Days
10	0.05	0.78	0.46	0.45
10	0.10	0.79	0.49	0.48
10	0.15	0.78	0.49	0.48
10	0.20	0.80	0.50	0.49
10	0.25	0.80	0.51	0.50
10	0.30	0.83	0.50	0.49
10	0.35	0.78	0.49	0.48
10	0.40	0.74	0.44	0.44

**Table 9 entropy-23-00731-t009:** Different weights of features and related experimental results.

Weight of 8 Features	1 Day	2 Days	3 Days
[0.3,0.1,0.1,0.1,0.1,0.1,0.1,0.1]	0.77	0.46	0.45
[0.1,0.3,0.1,0.1,0.1,0.1,0.1,0.1]	0.78	0.45	0.45
[0.1,0.1,0.3,0.1,0.1,0.1,0.1,0.1]	0.83	0.50	0.49
[0.1,0.1,0.1,0.3,0.1,0.1,0.1,0.1]	0.79	0.44	0.43
[0.1,0.1,0.1,0.1,0.3,0.1,0.1,0.1]	0.79	0.47	0.46
[0.1,0.1,0.1,0.1,0.1,0.3,0.1,0.1]	0.77	0.47	0.46
[0.1,0.1,0.1,0.1,0.1,0.1,0.3,0.1]	0.77	0.45	0.44
[0.1,0.1,0.1,0.1,0.1,0.1,0.1,0.3]	0.78	0.45	0.44

**Table 10 entropy-23-00731-t010:** Different variables’ values for different features.

Feature	Value of *δ*	Value of *n_ef_*
close	0.05	35
turnover rate	0.05	10
volume ratio	0.05	60
pe	0.1	30
pe_ttm	0.1	30
pb	0.005	50
ps	0.005	25
ps_ttm	0.005	24

**Table 11 entropy-23-00731-t011:** Comparison results of Group 2.

Type	Top 50_Rate	Top 100_Rate	Top 150_Rate
composite similarity	0.96	0.9	0.873
number of days rising or falling together	0.86	0.85	0.86
DTW(close)	0.92	0.89	0.86
average rate of whole sample	0.866		

**Table 12 entropy-23-00731-t012:** Structure of daily weather data.

Time	Aqi	PM 2_5	PM 10	CO	NO_2_	(O_3_, SO_2_, Temp)	Humi
1 January 2020	62	35	56	0.8	49	…	36.583
2 January 2020	80	51	80	1.2	64	…	41.875
3 January 2020	82	50	72	1.2	65	…	46.750
4 January 2020	74	43	66	1.1	59	…	44.542
5 January 2020	83	61	73	1.3	66	…	70.958
…	…	…	…	…	…	…	…
31 January 2021	89	66	83	1.1	34	…	77.269

**Table 13 entropy-23-00731-t013:** Chosen temporal features and their meanings.

Feature	Meaning
time	time index
pm2_5	Particulate Matter 2.5
pm10	Particulate Matter 10
no2	Nitrogen Dioxide
co	Carbon Oxide
o3	Ozone
so2	Sulfur Dioxide
temp	Temperature
humi	Humidity

**Table 14 entropy-23-00731-t014:** Top 10 similar cities and composite similarities.

No.	City	Composite Similarity
1	Langfang	2.8641
2	Zhangjiakou	1.6770
3	Baoding	1.5155
4	Tianjin	1.4962
5	Hengshui	1.4928
6	Dalian	1.4920
7	Dongying	1.4423
8	Chengde	1.3974
9	Qingdao	1.3807

**Table 15 entropy-23-00731-t015:** The rising-or-falling-together rate in top 50 similar cities.

Type	Top 50_Rate
composite similarity	0.880
number of days rising or falling together	0.700
DTW(temperature)	0.860
average rate of whole sample	0.862

## Data Availability

Data supporting reported results can be found at Tushare Pro (https://tushare.pro/, accessed on 1 February 2021) and AkShare (https://akshare.xyz/, accessed on 24 April 2021).

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
