# Peer review of "A Novel Time-Sensitive Composite Similarity Model for Multivariate Time-Series Correlation Analysis"

_entropy, 2021, doi:10.3390/e23060731_

Round 1
Reviewer 1 Report
The authors propose a time-sensitive composite similarity model designed for multivariate time-series correlation analysis based on dynamic time warping. Even if the proposed model will be reviewed for its validity, it must initially be suitable for application to (at least) a wider set of timeseries and not only to a specific form of timeseries. Instead of this, authors propose a similarity model that seems suitable for stock timeseries. I believe that the current manuscript is not suitable for publication in current journal since it is highly focused in a very specific subject without providing valuable or valid information of how this model will (or can) be used to other timeseries.
If the authors intend to focus only in stock timeseries then tha manuscript must be submitted to other journal
If the authors intend to provide a general model that may fit to the wider interest of journal readers with an example from stock timeseries then the current manuscript is propblematic due to the following reasons:
1) In the theoritical presentation of the method there is no clear indication of what is the contribution of the authors over the existing models. All the presented algorithms are authors' contribution or they rely on several existong ones and they just extend them?
2) there is no simulation with widely accepted control (reference) data sets along with comparison with existing or similar approaches. How the reader will be convinced if the authors were not provide results with widely accepted reference data sets. In other words, general evaluation of proposed model is missing. The used dataset (tushare.pro) cannot be characterized as reference dataset
3) Does the proposed model is useful for something else except for specific stocks' timeseries. If yes authors must provide details and evidence
I strongly believe that the Entropy journal is not the suitable one for this manuscript and the authors must submit their manuscript to a financial journal.
Reviewer 2 Report
I appreciate the work. Everything seems clear and easy to read. However, I would like to see more information about picking the optimal value of δ, since in the paper it is not so clear how to define the exact value one should use. May be it depends on some characteristics of the analysed time-series.
Reviewer 3 Report
In this paper, a novel method based on extending dynamic time warping is proposed to explore correlation between multivariate time series. The method is innovative and shows good performance empirically, especially compared to existing strategies such as 'roftrate'. The datasets that are used in the empirical study is of high relevance currently and questions addressed are of importance for similar datasets from everywhere in the world. The good performance of the new method in the empirical study is a very strong point of the paper.
Although the content of the paper is great amd the method novel, the presentation requires substantial improvement before publication. Here I point out several issues. However, I suggest the authors weigh different aspects of the presentation carefully with a potential reader in mind, and make thorough revisions regarding presentation, perhaps taking into consideration more than issues that I list below. Also, it'd be nice if the paper is a bit more self-contained too.
Detailed comments:
In section 2.1, a sentence about the following recently published paper would be of relevance:
Thomakos, D.; Klepsch, J.; Politis, D.N. Model Free Inference on Multivariate Time Series with Conditional Correlations. Stats 2020, 3, 484-509.
Since 'Dynamic Time Warping' is the key component of the method proposed in this paper, more details are necessary in section 2.3 to make it self-contained. For example, explain why Dynamic Time Warping will work. Are there any theoretical guarantees in the literature? Please explain more the intution behind boundary conditions, slope constraints, warping windows etc. before using them in the definition.
The notation 'n' is used to mean different things in different contexts, which can be confusing. In line-191, 'n' is the length of the time series, in line-278, 'n' stands for the # of features, in line-338, 'n' has yet another meaning. In line-346, 'n' means the number of eligible peaks and troughs.
The parameters 'delta' and 'lambdas' were chosen in the paper based on subjective considerations (which is fine in this introductory paper), but it'd be worth mentioning whether any objective criteria like cross-validation could be used for selecting them in future. Also, it'd be illuminating for the reader if the subjective reasoning behind choosing the particular set of weights in line-350 is explained. What was the intuition behind up-weighting only one stock? Also, for a practitioner, any guidance on the optimal # of features to be used? Also, the # of eligible peaks and troughs chosen for the emprical study was 50. Any guidance on how to choose this number in general? Would it depend on the length of the time series itself? Also any comment about over-fitting? - e.g. >8 features may improve the performance for the given dataset, but the performance may not be generalized to other external datasets?
Could other functional forms be considered in future, for the inverse relationship between similarity and DTW distance? (e.g. are there is any advantage in considering non-linear functions, or machine learning techniques?)
In lines 140 and 141, the words 'I' and 'my' makes it look more like an internal communication among the co-authors. If it was really intended for the external reader, please change it to 'we'.
How is the 'distance' calculated in lines 156-159?
Round 2
Reviewer 1 Report
The authors provide fair responses to the points i raised.
the manuscript (including obviously responses to other reviewers) is quite improved.
If the Editors find the current manuscript relevant to the scope of the journal then i suggest publication in its current form.